# COALA: Numerically Stable and Efficient Framework for Context-Aware Low-Rank Approximation

**Uliana Parkina**
HSE University
uliana.parkina@gmail.com

**Maxim Rakhuba**
HSE University

## Abstract

Recent studies suggest that context-aware low-rank approximation is a useful tool for compression and fine-tuning of modern large-scale neural networks. In this type of approximation, a norm is weighted by a matrix of input activations, significantly improving metrics over the unweighted case. Nevertheless, existing methods for neural networks suffer from numerical instabilities due to their reliance on classical formulas involving explicit Gram matrix computation and their subsequent inversion. We demonstrate that this can degrade the approximation quality or cause numerically singular matrices.

To address these limitations, we propose a novel *inversion-free regularized framework* that is based entirely on stable decompositions and overcomes the numerical pitfalls of prior art. Our method can handle possible challenging scenarios: (1) when calibration matrices exceed GPU memory capacity, (2) when input activation matrices are nearly singular, and even (3) when insufficient data prevents unique approximation. For the latter, we prove that our solution converges to a desired approximation and derive explicit error bounds.

## 1 Introduction

Large Language Models (LLMs) have demonstrated high performance across a variety of tasks [52], leading to significant advancements in artificial intelligence. However, the increasing size of these models brings efficiency challenges, including inference speed and model size when resources are constrained [55, 41, 6, 54]. To address these issues, various approaches have been proposed, such as model compression and fine-tuning. Specifically, basic and context-aware low-rank approximation techniques have proven to be an effective tool for compressing [46, 50, 25, 6, 27, 21] and fine-tuning [33, 47, 42, 43] modern large-scale neural networks.

In the context-aware approach, we consider the task of approximating a weight matrix $W \in \mathbb{R}^{m \times n}$ given input data $X \in \mathbb{R}^{n \times k}$, where $k$ is the batch size multiplied by the context length. The goal is to find a low-rank approximation $W'$ of $W$ that maintains the performance of the neural network while reducing its computational complexity. This objective leads to the minimization of

$$L(W') = \|WX - W'X\|_F,$$

where $\| \cdot \|_F$ denotes the Frobenius norm. We aim to minimize $L(W')$ with respect to the matrix $W' \in \mathbb{R}^{m \times n}$, under the constraint that the rank of $W'$ does not exceed $r$.

Despite its seeming simplicity, the context-aware low-rank approximation poses various challenges from the computational point of view:

**Numerical instabilities.** The first source of difficulty stems from numerical instability. Prior methods [46, 25, 47, 6] frequently depend on the inversion of large or nearly singular Gram matrices

39th Conference on Neural Information Processing Systems (NeurIPS 2025).

of the columns of $X$, which can degrade performance and introduce substantial computational errors in practice [44, 27]. Although theoretical guarantees for such inversion-based strategies often assume that the Gram matrix is of full rank, this condition can fail under real-world constraints and floating-point arithmetic [44, 27].

**Large calibration datasets.** Another challenge relates to memory limitations, particularly noticeable in large-scale scenarios. For instance, calibrating LLaMA3-8B [15] using 100 examples of length 2048 tokens and an internal dimensionality of 14336 leads a $\approx 10.9Gb$ matrix $X$ in a single precision. Thus, the method should be memory-efficient and, if necessary, support batch processing without explicitly constructing the whole matrix $X$.

**Limited data.** A final challenge emerges when dealing with severely limited data. In low-data regimes (e.g., 3–5 images in generative model adaptation [17, 14, 37]), the problem becomes ill-posed and susceptible to non-unique solutions and overfitting. Similar difficulties appear in model compression tasks when only constrained datasets are available [3, 29].

In this paper, we overcome all these issues in a single framework, called *COALA* (**CO**ntext-**A**ware **L**ow-rank **A**pproximation). Our contributions are as follows.

- We propose to use a regularization term that balances fitting the available examples with preserving the model's capacity to generalize. This regularized formulation yields unique solution for any $X$. Moreover, it boosts metrics of compressed neural networks by mitigating overfitting. Under certain assumptions, we establish a theoretical convergence of the regularized solution.

- We show how to fully avoid both inversion and computation of Gram matrices, which are prone to numerical instabilities [12]. Also, to avoid computational challenges associated with large matrices $X$, we preprocess them via the reliable TSQR algorithm [11], which computes a QR decomposition in smaller chunks.

## 2 Related Work

In recent years, the compression of deep learning models has become a critical focus within the field. Several approaches have been proposed to address this challenge, including quantization [48, 24], structured pruning [30, 2, 53], and low-rank approximation methods [22, 49, 45]. Quantization allows for reducing the bit-width of model weights, thereby decreasing memory consumption and accelerating computations. Structured pruning removes unnecessary parameters and simplifies the model architecture without significant loss of accuracy. Also, the low rank decomposition approach is often memory-efficient and can accelerate model inference, which is particularly important for tasks related to response speed and model size when deploying on mobile devices [50, 6, 28, 39, 7, 55].

The primary concept behind model compression using low-rank approximations is to represent the weight matrix $W$ as the product of two low-rank matrices: $W = UV$, where $W \in \mathbb{R}^{m \times n}, U \in \mathbb{R}^{m \times r}, V \in \mathbb{R}^{r \times n}$. This representation allows for storing $\mathcal{O}(mr + nr)$ elements instead of the original $\mathcal{O}(mn)$, and enables the propagation of data $X \in \mathbb{R}^{n \times k}$ through a layer with computational complexity $\mathcal{O}(nkr + mkr)$ rather than $\mathcal{O}(mnk)$, which is advantageous when $r \ll m, n$.

In this context, the theory of low-rank matrix approximations is developed to preserve certain properties of the original matrix. For instance, the Eckart–Young-Mirsky theorem [13], based on the singular value decomposition (SVD) [13] of a matrix, allows for the construction of a low-rank matrix $W'$ that best approximates $W$ in the sense of minimizing the norm $\|W - W'\|$ for any unitarily invariant norm, such as the Frobenius norm. However, as demonstrated in works [46, 50], such approximations do not always efficiently preserve the model's performance and are outperformed by compression methods based on alternative ideas, such as quantization, structured pruning, and unstructured pruning.

The work ASVD [50] proposes a solution that manages activation outliers by transforming the weight matrix based on the activation distribution. However, this solution does not achieve the best approximation error in the posed problem, providing a reasonable yet suboptimal solution [46]. Other studies, such as [46, 44], [25] and [6], present solutions that attain the theoretical minimum of the

error in the Frobenius norm. Nonetheless, they still rely on the formation of Gram matrices and/or inversion of small singular values.

## 3   Weighted Low-Rank Approximation

Building upon the previous discussion, by introducing a *context-aware* approach at each layer using $X$, we aim to reduce the number of parameters needed to store the matrix $W$ by finding its low-rank approximation. Formally, this can be formulated as the following minimization problem:

$$\min_{\text{rank}(W') \leq r} \|(W - W')X\|_F. \tag{1}$$

This problem is a special case of the general weighted low-rank approximation problem [31, 32]:

$$\min_{\text{rank}(W') \leq r} \text{vec}\{W - W'\}^\top Q \, \text{vec}\{W - W'\},$$

where $Q$ is positive definite symmetric matrix and $\text{vec}\{\cdot\}$ denotes the column-wise vectorization of a matrix. By applying [31, Theorem 3] (see Appendix A) in our specific case of the matrix $Q$, we obtain:

$$W' = U\Sigma_r V^\top S^{-1}, \tag{2}$$

where $S = (XX^\top)^{1/2}$ is the unique positive definite square root of $XX^\top$, $U\Sigma V^\top$ is the SVD of $WS$, and $\Sigma_r$ is the matrix obtained from $\Sigma$ by setting the last $n - r$ singular values to zero.

One can show that using the symmetric matrix square root is not the only possible way to obtain the solution. Any decomposition of the form $SS^\top = XX^\top$ with a square matrix $S$ is applicable as well. For example, $S$ can be an $R^\top$ factor from the Cholesky decomposition of the matrix $XX^\top$. Alternatively, it can be based on SVD of $XX^\top$. For example, these two approaches were used in [44, 46] and utilized in other studies [25, 47], see Appendix B. As we will see further, forming the Gram matrix in this context may already lead to numerical problems in the ill-conditioned case, see also a theoretical example from Appendix G.1. Inverting nearly singular matrices afterwards only deteriorate this effect. In the next section, we show how to naturally avoid both problems at once.

## 4   Inversion-Free Solution

The following result provides a simple yet effective orthogonal-projection-based formula, avoiding matrix inversion and Gram matrices.

**Proposition 1.** *Let $W \in \mathbb{R}^{m \times n}$ and $X \in \mathbb{R}^{n \times k}$ be arbitrary matrices. A solution to the optimization problem*

$$\min_{\text{rank}(W') \leq r} \|WX - W'X\|_F \tag{3}$$

*is attained at $W' = U_r U_r^\top W$, where $U_r$ consists of the first $r$ left singular vectors of the matrix $WX$.*

*Proof.* Let us define $A = WX$ and $B = W'X$. Then,

$$\text{rank}(B) \leq \min\left(\text{rank}(W'), \text{rank}(X)\right) \leq \text{rank}(W') \leq r.$$

It is well-known that the minimizer of $\|A - B\|_F$ under the constraint $\text{rank}(B) \leq r$ is given by $B = U_r U_r^\top A$, where $U_r$ contains the first $k$ left singular vectors of $A$ (see Corollary 2 for details). Substituting back for $A$ and $B$, we have $B = U_r U_r^\top A = U_r U_r^\top WX$, implying

$$W'X = U_r U_r^\top WX.$$

Hence, one of the possible solutions $W'$ looks as follows:

$$W' = U_r U_r^\top W.$$

As desired, the rank of $W'$ does not exceed $r$ because $U_r$ has rank $r$. Note that in general there can be many solutions depending on the matrix $X$. $\qquad \square$

Although this result is well-established for the unweighted case ($X = I$), we include a proof here since we were unable to find a weighted analogue in the literature. Note that this formula does not require any additional constraints on $X$, such as the assumption of full column rank, which is required in [46, 44]. However, let us note that the number of columns $k$ of the matrix $X$ grows with the number of samples and can be a fairly large quantity, exceeding $m$ and $n$ by many times. Nevertheless, it can can be efficiently computed with the help of the reliable QR decomposition.

**Proposition 2.** *Suppose that $n \leq k$. Then, we can get $U_r$ in Proposition 1 as the first $r$ left singular vectors of the matrix $WR^\top$, where $R$ is the upper triangular matrix from the QR decomposition of $X^\top$.*

*Proof.* Let $QR = X^\top$ be the QR decomposition of $X^\top$. Then, using orthogonal invariance of $\| \cdot \|_F$:

$$\|(W' - W)X\|_F^2 = \|(W' - W)R^\top Q^\top\|_F^2 =$$
$$= \operatorname{tr}\left((W' - W)R^\top Q^\top Q R(W' - W)^\top\right) = \left[Q^\top Q = I\right] =$$
$$= \operatorname{tr}\left((W' - W)R^\top R(W' - W)^\top\right) = \|(W' - W)R^\top\|_F^2.$$

We complete the proof by applying Proposition 1 to the new minimization task. $\square$

Note that in the proof of Proposition 2, we only use the fact that $R^T R = X X^T$, so any matrix for which this is true will suffice.

The pseudocode of the final solution is summarized in Algorithm 1.

---

**Algorithm 1** A Stable Solution to the Weighted Low-Rank Approximation Problem

---

**Require:** $W \in \mathbb{R}^{m \times n}, X \in \mathbb{R}^{n \times k}, r \in \mathbb{N}, n \leq k$
**Ensure:** $A \in \mathbb{R}^{m \times r}, B \in \mathbb{R}^{r \times n}$
  1 **Compute** the upper–triangular factor $R$ by performing a TSQR factorization of $X^\top$:
     $[Q, R] \leftarrow \mathrm{QR}(X^\top)$                    ▷ Use the Tall–Skinny QR (TSQR) method, see Section 4.2.
  2 **Compute** the SVD of $WR^\top$:
     $[U, \Sigma, V^\top] \leftarrow \mathrm{SVD}(WR^\top)$
  3 **Let** $U_r = U[:, : r]$
  4 **Set** $A \leftarrow U_r$
  5 **Set** $B \leftarrow U_r^\top W$
  6 **return** $A, B$

---

## 4.1 Stability

Let us analyze potential issues that arise on real-world data. In particular, we use the LLaMA3 [15] model on the WikiText2 [34] dataset and construct weighted low-rank approximation of matrices in three ways: (1) via Cholesky decomposition of $(XX^\top)$ as in SVD-LLM, (2) via $\mathrm{SVD}$ of $(XX^\top)$ as in SVD-LLM v2, and (3) via the QR-based approach.

Figure 1 shows that the approaches relying on the Gram matrix suffer from large errors that are independent of the chosen rank and appear already during the construction of the approximation of $W$. We evaluate the error in the spectral norm $\| \cdot \|_2$, the operator norm induced by the Euclidean vector norm via $\|A\|_2 = \sup_{x \neq 0} \|Ax\|_2 / \|x\|_2$. Being defined through a supremum over all possible inputs, this bound cannot be exceeded by any particular vector $x$.

The size of these errors is linked to the distribution of singular values: very small singular values cause numerical instabilities when inversion of the Gram matrix is involved. As illustrated in Figure 2, several layers exhibit a sharp drop in the smallest singular values of the input matrix $X$. Our findings indicate that computing $XX^\top$ introduces noticeable numerical errors, which may subsequently impact the final results.

## 4.2 Efficiency

In this section, we also discuss the compression time for large models. In our approach we preprocess $X$ using the QR decomposition, see Proposition 2. The need for only the $R$ factor in the QR decomposition provides further acceleration.

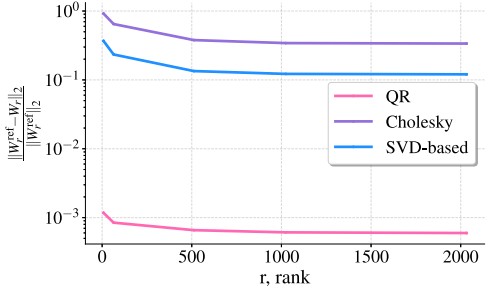

Figure 1: Relative approximation error versus approximation rank obtained by different methods on layer 1 `q_proj`. The reference weight matrix $W_r^{\text{ref}}$ was computed using the inversion-free COALA method and in high working precision (fp64) to serve as the ground-truth solution. The LLaMA3-1B [15] model was used with 64 examples from the Wikitext [34] dataset.

Figure 2: Distribution of singular values of matrix $X$, obtained from the outputs of layer 1 `q_proj` in the LLaMA3-1B [15] model, computed over 64 samples from the WikiText [34] dataset.

Table 1: Computation times produced by different methods.

| Model | #Samples | Strategy | Time, s |
|-------|----------|----------|---------|
| LLaMA3-1B | 64 | SVD-LLM | $\underline{273.93}\pm22.12$ |
| | | SVD-LLM V2 | $404.88\pm5.49$ |
| | | COALA | $\mathbf{196.34}\pm6.48$ |
| LLaMA3-8B | 128 | SVD-LLM | $\underline{3624.88}\pm512.4$ |
| | | SVD-LLM V2 | $4210.5\pm63.3$ |
| | | COALA | $\mathbf{1811.0}\pm15.6$ |

We compared the time required by different methods in Table 1. Additionally, we examined the breaking point at which computing the SVD of $XX^\top$ becomes faster than performing a QR decomposition of $X$. We have observed that even when the matrix has a highly unbalanced aspect ratio – with one dimension exceeding the other by several tens of times – the QR decomposition remains the preferred method, see Figure 3, left graph. All calculations were performed on a single NVIDIA A100 GPU. To preserve the integrity of the experiment, the SVD on the GPU was executed with PyTorch's "gesvd" method, because the default "gesvdj" method, although faster, produces a noticeably larger error.

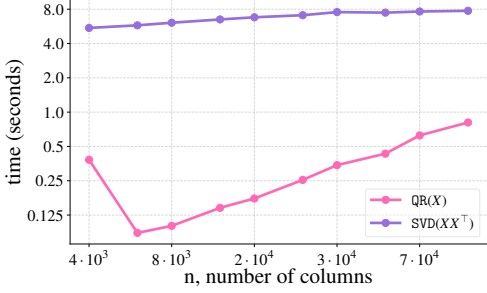
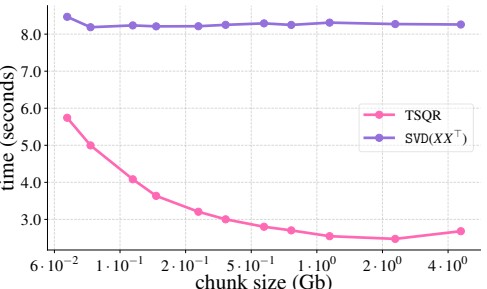

Figure 3: Runtimes for computing $S$: $SS^\top = XX^\top$ using two approaches. *Left:* Matrix $X \in \mathbb{R}^{4096\times n}$ for different $n$. *Right:* Matrix $X \in \mathbb{R}^{4096\times 3\cdot 10^5}$ split into chunks of different size. In this case, QR is computed using the TSQR method and the Gram matrix using $XX^\top = \sum_{i=1}^{p} X_i X_i^\top$.

When dealing with matrices $X$ so large that they cannot be accommodated in fast memory, one can still easily compute the Gram matrix by splitting it into $p$ batches of smaller sizes that fit in

memory, resulting into $XX^\top = \sum_{i=1}^{p} X_i X_i^\top$. In our case, we can also efficiently compute the QR decomposition using the Tall Skinny QR (TSQR) method [11]. It allows for reducing QR decomposition of the whole matrix to $p$ QR decompositions of the smaller sizes. For example, we can sequentially apply the QR decomposition to each new block, incorporating the $R$ matrix obtained from the previous step, i.e., for $p = 3$:

$$X^\top = \begin{bmatrix} X_0^\top \\ X_1^\top \\ X_2^\top \end{bmatrix} = \begin{bmatrix} Q_0 R_0 \\ X_1^\top \\ X_2^\top \end{bmatrix} = \begin{bmatrix} Q_0 & & \\ & I & \\ & & I \end{bmatrix} \begin{bmatrix} R_0 \\ X_1^\top \\ X_2^\top \end{bmatrix} =$$

$$= \begin{bmatrix} Q_0 & & \\ & I & \\ & & I \end{bmatrix} \begin{bmatrix} Q_{01} & \\ & I \end{bmatrix} \begin{bmatrix} R_{01} \\ X_2^\top \end{bmatrix} = \begin{bmatrix} Q_0 & & \\ & I & \\ & & I \end{bmatrix} \begin{bmatrix} Q_{01} & \\ & I \end{bmatrix} Q_{012} R_{012}.$$

Since a product of matrices with orthonormal columns also has orthonornal columns, we conclude that this is indeed a QR decomposition of $X^\top$. As can be seen in Figure 3 (right), this approach not only eliminates the need to store large matrices, but also speeds up the solution time for large-scale matrices $X$.

Moreover, if multiple GPUs are available, the scheme can be transformed into a binary tree structure to enable parallel execution, thereby achieving a speedup in computational time:

$$\begin{array}{cccc}
X_0 \to R_0 & \searrow & & \\
 & & R_{01} & \searrow \\
X_1 \to R_1 & \nearrow & & \\
 & & & R_{0123} \\
X_2 \to R_2 & \searrow & & \nearrow \\
 & & R_{23} & \\
X_3 \to R_3 & \nearrow & &
\end{array}$$

If one or more arrows point to the same matrix, then this matrix represents the R-factor obtained from the QR decomposition of the matrix formed by stacking all the matrices at the opposite ends of the arrows. This process is described in more detail in [11].

## 5   Weighted Low-Rank Approximation with Regularization

So far, we have discussed various numerical aspects of solving the problem (3). However, in practice, we want to adapt the model to fit the available examples, but not excessively, as we aim to avoid overfitting and preserve the model's knowledge in other domains. This situation becomes particularly pronounced when data is scarce and matrix $X$ may have more columns than rows. For example, in model compression, data are often limited due to confidentiality, yet there's a need to deploy models on devices with restricted resources [3, 29]. A similarly relevant challenge is adapting pre-trained generative models to new concepts using just a handful of images (usually 3–5) [17]. Thus, we can formulate the following minimization problem:

$$\min_{\text{rank}(W') \leq k} \|WX - W'X\|_F^2 + \mu\|W - W'\|_F^2, \tag{4}$$

where $\mu \geq 0$ is a given parameter. Notably, this strategy yields systematic improvements even in data-sufficient scenarios.

Our methodology presented in earlier sections continues to provide an efficient and robust solution to the problem (4) as well.

**Proposition 3.** *Let $W \in \mathbb{R}^{m \times n}$ and $X \in \mathbb{R}^{n \times k}$ be arbitrary matrices. Then problem (4) is equivalent to*

$$\min_{\text{rank}(W') \leq k} \|(W - W')\widetilde{X}\|_F^2,$$

*where $\widetilde{X} = [X \quad \sqrt{\mu}I]$.*

*Proof.* We have

$$\|(W' - W)X\|_F^2 + \mu\|W' - W\|_F^2 = \| [(W' - W)X \quad \sqrt{\mu}(W' - W)] \|_F^2 =$$
$$= \|(W' - W) \cdot [X \quad \sqrt{\mu} \cdot I] \|_F^2.$$

$\square$

This equivalence means that we can use the same approach as in the unregularized problem by augmenting the data matrix X with the scaled identity matrix $\sqrt{\mu}I$. By transforming the regularized problem into this form, we can apply efficient algorithms such as Proposition 2 for its solution. The corresponding pseudocode is presented in Algorithm 2.

---

**Algorithm 2** A solution to the weighted low-rank approximation problem with regularization

---

**Require:** $W \in \mathbb{R}^{m \times n}$, $X \in \mathbb{R}^{n \times k}$, $\mu \in \mathbb{R}_+$, $r \in \mathbb{N}$
**Ensure:** $A \in \mathbb{R}^{m \times r}$, $B \in \mathbb{R}^{r \times n}$
  1 **Form** the matrix $X' = [X \quad \sqrt{\mu}\,I]$, where $I$ is the $n \times n$ identity matrix.
  2 **Call** Algorithm 1 with input $(W, X', r)$ to compute $A$ and $B$.
  3 **return** $A, B$

---

**What is the limit of $W_\mu$ as $\mu \to 0$?**   Another natural question is what happens with the regularized solution $W_\mu$ for small $\mu$. If $X$ is of full row rank, then it is natural to assume that it $W_\mu$ converges to a unique solution of the unregularized problem. It is, however, unclear what happens in the general case and what is the convergence rate. We establish that $W_\mu$ converges to a well-defined solution $W_0$, which corresponds to the solution obtained from Proposition 1. The following theorem provides a precise estimate for the convergence rate.

**Theorem 1.** *Let $W \in \mathbb{R}^{m \times n}$ and $X \in \mathbb{R}^{n \times k}$. Suppose that $X$ has $\mathrm{rank}(X) = k \geq r$ and that the singular values of $WX$ satisfy $\sigma_r(WX) \neq \sigma_{r+1}(WX)$, where $\sigma_i(\cdot)$ denotes the $i$-th largest singular value. Let $W_0 = U_r U_r^\top W$ denote the solution to the problem* (3), *and let $W_\mu$ denote the solution to the regularized problem* (4). *then the following estimate holds:*

$$\|W_0 - W_\mu\|_F \leq \frac{2\|W\|_2^2 \|W\|_F}{\sigma_r^2(WX) - \sigma_{r+1}^2(WX)} \cdot \mu.$$

*Proof.* See Appendix E. □

In the case where $X$ has full rank, we have a more precise estimate with a better constant, which, however, also involves the multiplier $1/\mathrm{gap}$, where

$$\mathtt{gap} = \sigma_r(WX) - \sigma_{r+1}(WX),$$

see Appendix D. This gap-dependent behavior is intrinsic to the problem, as demonstrated in Example G.2.

Our estimate suggests that even in the degenerate case $W_\mu$ approaches $W_0$ linearly with respect to $\mu$ as $\mu \to 0$, which we also observe in numerical simulations. The estimate may also be useful for practical reasons as it shows asymptotic dependence on key parameters such as the gap value and the regularization parameter $\mu$. For example, the estimate quantifies how sensitive our solution is to the choice of $\mu$, which can inform practical decisions about selecting an appropriate regularization parameter. This can be of particular interest in applications where the balance between fitting the data and preventing overfitting is delicate.

## 6 Experiments

### 6.1 Model compression

In this section, we evaluate the effectiveness of our regularization-based compression approach in practice [1]. We first fix the procedure for selecting the regularization parameter $\mu$. Specifically, we determine $\mu$ relative to the unregularized solution $W_0$ (i.e., the one obtained for $\mu = 0$) according to the formula below:

$$\mu = \frac{\|W_0 X - WX\|_F^2}{\|W_0 - W\|_F^2} \cdot \lambda, \tag{5}$$

where $\lambda$ serves as a hyperparameter controlling the adjustment. This step is crucial because different layers of large language models exhibit substantially different norms of the weight matrices $W$,

---

[1]Our code is available at `https://github.com/urparkina/COALA`.

calibration matrices $X$, and their products $WX$, see, e.g., [19]. Moreover, we compare these two strategies, analyzing how the metric depends on adaptive and non-adaptive choices of $\mu$ across layers, see Figure 4. The Mistral-7B-Instruct model was selected due to its pronounced variation of layer-wise norms.

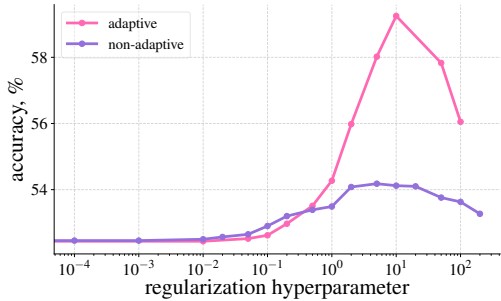

Figure 4: Comparison of the impact of parameter tuning with (Equation (5)) and without considering layer-wise norms on model quality at 70% compression, evaluated on a common-sense reasoning dataset using the *Mistral-7B-Instruct* model.

Figure 5 presents sensitivity analysis of the parameter $\lambda$, demonstrating that the optimal value of $\mu$ remains relatively stable (in the region from 1 to 10) across different settings, including various model architectures, datasets, and compression ratios.

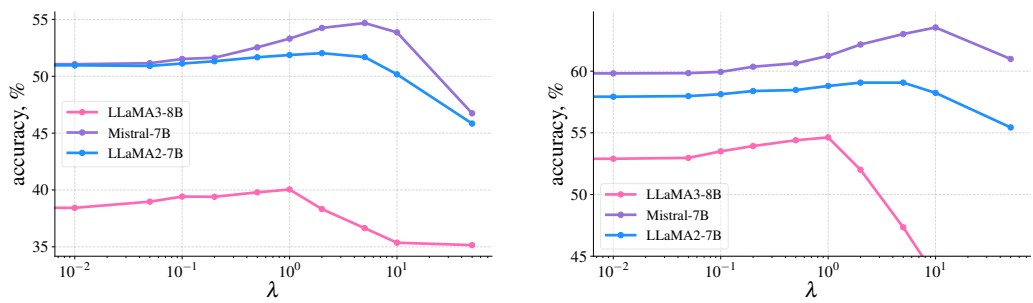

Figure 5: The dependence of average accuracy on the parameter $\lambda$ on the commonsense reasoning dataset for different models. On the left: 70% compression ratio, on the right: 80% compression ratio.

Table 2 compares several methods without adaptive rank selection under reduced-precision (fp16) conditions. We observe that our more numerically stable formulations improve performance, with regularization providing the most consistent gains. We compare our method to approaches that do not use fine-tuning or adaptive rank selection. However, our solution can be potentially used not only as a standalone compression technique, but also integrated into other works as a part of a problem-solving framework.

Finally, Table 3 compares our approach with state-of-the-art methods that report the relevant metrics in their manuscripts. Our method achieves comparable or even superior results solely due to the use of regularization, without any additional heuristics or fine-tuning.

Table 2: Metric values of various compression methods. All computations, except for solving the weighted low-rank approximation problem, were performed in half precision (fp16). Experiments were conducted using the *LLaMA-3.2-1B-Instruct* model compressed at 90% using text samples from the commonsense reasoning dataset, which was also used for validation.

| Method | boolQ | PIQA | WiNoG | HSwag | ARC-E | ARC-C | OBQA |
|---|---|---|---|---|---|---|---|
| Original | $69.5_{\pm 0.7}$ | $74.4_{\pm 1.0}$ | $59.5_{\pm 1.3}$ | $60.7_{\pm 0.5}$ | $63.2_{\pm 0.9}$ | $38.1_{\pm 2.1}$ | $34.6_{\pm 1.4}$ |
| ASVD | $\mathbf{58.0}_{\pm \mathbf{0.7}}$ | $52.5_{\pm 1.0}$ | $51.3_{\pm 1.3}$ | $27.8_{\pm 0.5}$ | $30.0_{\pm 0.9}$ | $25.9_{\pm 2.1}$ | $\underline{26.8}_{\pm 1.4}$ |
| SVD-LLM | $54.1_{\pm 0.7}$ | $60.6_{\pm 1.0}$ | $53.8_{\pm 1.3}$ | $34.6_{\pm 0.5}$ | $\underline{44.3}_{\pm 0.9}$ | $25.5_{\pm 2.1}$ | $26.0_{\pm 1.4}$ |
| COALA$_{\mu=0}$ | $57.6_{\pm 0.7}$ | $60.9_{\pm 1.0}$ | $\underline{53.2}_{\pm 1.3}$ | $34.6_{\pm 0.5}$ | $43.4_{\pm 0.9}$ | $27.3_{\pm 2.1}$ | $26.0_{\pm 1.4}$ |
| COALA$_{\mu}$ | $59.0_{\pm 0.7}$ | $\mathbf{62.8}_{\pm \mathbf{1.0}}$ | $\mathbf{54.0}_{\pm \mathbf{1.3}}$ | $\mathbf{36.6}_{\pm \mathbf{0.5}}$ | $\mathbf{46.2}_{\pm \mathbf{0.9}}$ | $\mathbf{29.2}_{\pm \mathbf{2.1}}$ | $\mathbf{27.6}_{\pm \mathbf{1.4}}$ |

Table 3: Metric values of various compression methods. Experiments were conducted using the *Mistral-7B* model on the WikiText2 dataset and commonsense reasoning used for validation. The results for SliceGPT [2] and FLAP [1] were taken from the work [25].

| Ratio | Method | MMLU | BoolQ | PIQA | WiNoG | HSweg | ARC-E | ARC-C | OBQA |
|---|---|---|---|---|---|---|---|---|---|
| 100% | Mistral-7B | 62.50 | 83.98 | 82.05 | 73.95 | 81.02 | 79.55 | 53.92 | 44.00 |
| 80% | FLAP | 25.90 | 62.26 | 72.31 | 64.09 | 55.94 | 51.05 | 31.91 | 36.80 |
| | SliceGPT | 28.60 | 37.86 | 60.66 | 59.43 | 45.10 | 48.15 | 30.03 | 32.00 |
| | SVD-LLM | $\underline{41.80}$ | $\underline{68.29}$ | 73.39 | 68.43 | 61.75 | $\underline{71.34}$ | $\underline{40.53}$ | 36.60 |
| | SoLA | $\mathbf{44.20}$ | 66.09 | $\underline{73.67}$ | $\underline{68.75}$ | $\underline{63.32}$ | 69.99 | 39.76 | $\underline{39.20}$ |
| | COALA | 41.20 | $\mathbf{78.07}$ | $\mathbf{77.04}$ | $\mathbf{68.82}$ | $\mathbf{65.06}$ | $\mathbf{72.13}$ | $\mathbf{43.43}$ | $\mathbf{40.20}$ |
| 70% | FLAP | 26.40 | $\mathbf{65.26}$ | $\underline{69.59}$ | $\mathbf{64.80}$ | $\mathbf{55.61}$ | 48.91 | 30.55 | 35.80 |
| | SliceGPT | 25.00 | 37.83 | 54.41 | 51.62 | 32.54 | 35.02 | 22.95 | 26.80 |
| | SVD-LLM | $\underline{28.20}$ | $\underline{64.62}$ | 64.91 | 64.17 | 47.36 | 58.25 | 30.72 | 34.20 |
| | SoLA | $\mathbf{33.80}$ | 62.57 | 68.39 | $\underline{64.48}$ | $\underline{53.00}$ | $\underline{60.90}$ | $\underline{32.76}$ | $\mathbf{37.60}$ |
| | COALA | 27.35 | 63.82 | $\mathbf{70.40}$ | 62.43 | 51.02 | $\mathbf{63.63}$ | $\mathbf{35.49}$ | $\underline{36.00}$ |

We conducted experiments on the models LLaMA3-8B, LLaMA3-1B [16] and Mistral-7B [5] (including Insrtuct versions), comparing our approach with existing methods across various datasets: boolQ [8], OpenbookQA [35], WinoGrande [38], HellaSwag [51], Arc_e [9], Arc_c [10], PIQA [4], MMLU [18]. We used A100 GPU and Tesla T4 GPU for our experiments. The results indicate that in all the considered settings our regularized algorithm systematically achieves better metrics during compression.

## 6.2 Fine-Tuning

Table 4: Results of fine-tuning LLaMA3-1B-Instruct at rank $r = 8$ using different PEFT initialization methods on the commonsense reasoning dataset with 24 examples for initialization. In exact arithmetic, "COALA $\alpha = 2$" is equivalent to CorDA. See hyperparameters in Appendix F.

| Method | BoolQ | PIQA | SIQA | HSweg | WiNoG | ARC-e | ARC-c | OBQA | Avg. |
|---|---|---|---|---|---|---|---|---|---|
| LoRA | $\mathbf{64.5}$ | $\mathbf{76.1}$ | 71.5 | 82.4 | 53.8 | 76.8 | 58.5 | 68.2 | 75.0 |
| PiSSA | $\mathbf{64.5}$ | $\underline{76.0}$ | 71.5 | $\mathbf{83.0}$ | 52.0 | $\mathbf{78.4}$ | $\mathbf{60.8}$ | $\mathbf{70.4}$ | $\underline{75.4}$ |
| CorDA | 61.4 | 68.7 | 62.1 | 60.8 | 52.4 | 68.7 | 40.1 | 52.8 | 60.9 |
| COALA $\alpha = 2$ | $\underline{64.4}$ | 75.9 | $\underline{72.6}$ | 82.7 | $\underline{54.3}$ | $\underline{78.2}$ | 59.5 | 68.0 | $\underline{75.4}$ |
| COALA $\alpha = 1$ | 64.1 | $\mathbf{76.1}$ | $\mathbf{72.8}$ | $\underline{82.8}$ | $\mathbf{56.0}$ | 77.5 | $\underline{59.8}$ | $\underline{68.4}$ | $\mathbf{75.5}$ |

Training and fine-tuning models with specific constraints or regularization applied to the weights has proven to be an effective technique in recent years [23, 40]. Fine-tuning methods often utilizes the concept of low-rank matrix approximations for initialization, see PiSSA [33] and CorDA [47] appraoches. We investigate the application of our method for initializing LoRA [23] adapters and demonstrate its advantages. The following proposition unifies these methods and also leads to a new method for $\alpha = 1$.

**Proposition 4.** *The solution to the optimization problem*

$$\min_{\text{rank}(W') \leq r} \text{tr}\left((W - W')(XX^\top)^\alpha (W - W')^\top\right) \tag{6}$$

*for an arbitrary $\alpha \geq 0$, $\alpha \in \mathbb{Z}$, is given by the formula:*

$$W' = U_r U_r^\top W, \quad \text{where} \quad U\Sigma V^\top = W(XX^\top)^{\frac{\alpha}{2}}$$

*and $U_r$ consists of the first $r$ columns of the matrix $U$.*

*Proof.* Note, that

$$\text{tr}\left((W - W')(XX^\top)^\alpha (W - W')^\top\right) = \|(W - W')(XX^\top)^{\frac{\alpha}{2}}\|_F^2,$$

where $(XX^\top)^{\frac{\alpha}{2}} = S$ is such a square positive definite matrix that $SS^\top = (XX^\top)^\alpha$. Thus, applying Proposition 1 , we obtain the desired solution. $\qquad\square$

Note that to obtain $(XX^\top)^{\frac{\alpha}{2}}$ one does not have to compute $XX^\top$ explicitly. One possible strategy is to take the SVD of $X$: $X = U\Sigma V^\top$ and then $(XX^\top)^{\frac{\alpha}{2}} = U\Sigma^{\frac{\alpha}{2}} U^\top$.

**Remark 1.** *For $\alpha = 2$, the task* (6) *becomes equivalent to the following minimization problem:*

$$\min_{\text{rank}(W') \leq r} \text{tr}\left((W - W')(XX^\top)^2 (W - W')^\top\right) = \min_{\text{rank}(W') \leq r} \|(W - W')XX^\top\|_F^2.$$

*Thus, applying Corollary 1 , we arrive at the solution*

$$W' = U_r \Sigma_r V_r^\top (XX^\top)^{-1},$$

*where $U\Sigma V^\top = WXX^\top$ and $U_r, \Sigma_r, V_r^\top$ are truncated matrix. This solution is presented as an algorithm in the CorDA method.*

*By applying our Proposition 1 , we can obtain another way of solving this problem:*

$$W' = U_r U_r^\top W,$$

*where $U_r$ consists of the first $r$ left singular vectors of the matrix $WXX^\top$.*

We show that the solution provided by the CorDA method solves the problem described in (6), when $\alpha = 2$, and also applied our formulas for robustness purposes. Without them, in some scenarios, inversions of $XX^\top$ raised runtime errors due to singular matrices or lead to large numerical errors. Note also that for $\alpha = 0$ the minimization problem (6) leads to the PiSSA method. We conduct experiments on the LLaMA3-1B-Instruct [15] model. Table 4 suggests that the robustified version of CorDA (COALA, $\alpha = 2$) significantly boosts the performance. Both robust versions for $\alpha = 1$ and $\alpha = 2$ yield results similar to PiSSA, though $\alpha = 1$ performs slightly better.

## 7   Limitations

The limitations of our work are closely linked to the applicability and effectiveness of the weighted approximation approach. Thus, its efficiency is limited to tasks and domains where these methods perform well.

## 8   Conclusion

In conclusion, we have presented a new, regularized inversion-free framework for context-aware low-rank approximation of LLM. We aimed to address the issue of numerical instability seen in previous works, and developed solutions for challenging scenarios such as large calibration matrices exceeding GPU memory capacity and near-singular input activation matrices. In our experiments, we observed favorable results in both model compression and fine-tuning scenarios compared to previous methods.

## Acknowledgments

The work was supported by the grant for research centers in the field of AI provided by the Ministry of Economic Development of the Russian Federation in accordance with the agreement 000000C313925P4E0002 and the agreement with HSE University № 139-15-2025-009. The calculations were performed in part through the computational resources of HPC facilities at HSE University [26].

The authors are also grateful to A. Osinsky for insightful suggestions that led to an improved theoretical bound.

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

# A General Weighted Low-Rank Approximation Problem

**Definition A.1** (General Weighted Low-Rank Approximation Problem). *Given a matrix $W \in \mathbb{R}^{m \times n}$, we aim to find a matrix $W'$ of rank at most $r$, such that the objective function*

$$\min_{W':\text{rank}(W') \leq r} \text{vec}\{W - W'\}^{\top} Q \, \text{vec}\{W - W'\} \tag{7}$$

*is minimized, where $\text{vec}\{\cdot\}$ denotes the vectorization operator, transforming a given matrix into a column vector by stacking its columns on top of each other. The matrix $Q \in \mathbb{R}^{mn \times mn}$ represents a positive definite matrix.*

**Theorem 2.** *(From [31]) In (7), if $Q = Q_1 \otimes Q_2$, where $Q_1 \in \mathbb{R}^{m \times m}$ and $Q_2 \in \mathbb{R}^{n \times n}$ are both positive definite and symmetric, then the solution $W'$ of (7) is given by the following closed-form expression. Let $Q_2^{1/2} W Q_1^{1/2} = U \Sigma V^{\top}$ be the compact SVD, where $Q_1^{1/2}$ is the unique positive definite symmetric matrix such that $Q_1^{1/2} Q_1^{1/2} = Q_1$ and similarly for $Q_2^{1/2}$. Then, $W' = Q_2^{-1/2} U \Sigma_r V^{\top} Q_1^{-1/2}$, where $\Sigma_r$ is obtained from $\Sigma$ by setting all but the first $r$ singular values to zero. Here, $\otimes$ is the Kronecker product [20].*

*Proof.* See work [31]. $\square$

Observe that if we choose $Q_2 = I$ and $Q_1 = XX^{\top}$, we immediately obtain a solution to the problem (1) . More generally, note that any square matrix $S$ satisfying $SS^{\top} = XX^{\top}$ can be employed in this construction. For instance, a standard choice would be the Cholesky factor of $XX^{\top}$.

**Corollary 1.** *Let $W$ and $X$ be arbitrary matrices belonging to $\mathbb{R}^{m \times n}$ and $\mathbb{R}^{n \times k}$ respectively, with $X$ having full row rank.. The solution to the optimization problem (1) can be obtained using the formula*

$$W' = U \Sigma_r V^{\top} (XX^{\top})^{-1/2},$$

*where $U \Sigma V^{\top} = W(XX^{\top})^{1/2}$ is SVD.*

*Proof.* Note, that

$$\|(W - W')X\|_F = \text{tr}((W - W')XX^{\top}(W - W')^{\top}) = \left[\text{tr}(AB) = \text{vec}\{A\}^{\top} \text{vec}\{B^{\top}\}\right] =$$
$$= \text{vec}\{W - W'\}^{\top} \text{vec}\{(W - W')XX^{\top}\} =$$
$$= \text{vec}\{W - W'\}^{\top} \text{vec}\{I(W - W')XX^{\top}\} = \left[\text{vec}\{ABC\} = (C^{\top} \otimes A)\text{vec}\{B\}\right] =$$
$$= \text{vec}\{W - W'\}^{\top}(XX^{\top} \otimes I)\text{vec}\{W - W'\}.$$

So, if $X$ has a full rank, we can apply Theorem 2, where $Q_1 = XX^{\top}, Q_2 = I$:

$$W' = I^{-1/2} U \Sigma_r V^{\top} (XX^{\top})^{-1/2} = U \Sigma_r V^{\top} (XX^{\top})^{-1/2},$$

where $U \Sigma V^{\top} = W(XX^{\top})^{1/2}$. $\square$

# B SVD-LLM Method

In this section, we present pseudocode for several approaches to solve the problem, including the method outlined in Section A, an approach leveraging the Cholesky decomposition, and one utilizing the square root of the matrix $XX^{\top}$.

The Algorithm 3 from the work [46] provides the solution via Cholesky decomposition for the matrix $XX^{\top}$, while the Algorithm 4 from the work [44] finds the solution via the search for symmetric matrix square root of $XX^{\top}$ through SVD.

---

**Algorithm 3** SVD-LLM method [46]

---

    **Input:** $W \in \mathbb{R}^{m \times n}$, $X \in \mathbb{R}^{n \times k}$, $r \in \mathbb{N}$
    **Output:** $A \in \mathbb{R}^{m \times r}$, $B \in \mathbb{R}^{r \times n}$
1 **Compute** the upper triangular matrix $S$ from the Cholesky decomposition of $XX^\top$:
    $S \leftarrow \text{cholesky}(XX^\top)$
2 **Compute** the singular value decomposition of $WS$:
    $[U, \Sigma, V^\top] \leftarrow \text{svd}(WS)$
3 **Let** $U_r, \Sigma_r, V_r = U[:, :r], \Sigma[:r, :r], V_r[:, :r]$
4 **Set** $A \leftarrow U_r$
5 **Set** $B \leftarrow \Sigma_r V_r^\top S^{-1}$
6 **return** $A, B$

---

---

**Algorithm 4** SVD-LLM V2 method [44]

---

    **Input:** $W \in \mathbb{R}^{m \times n}$, $X \in \mathbb{R}^{n \times k}$, $r \in \mathbb{N}$
    **Output:** $A \in \mathbb{R}^{m \times r}$, $B \in \mathbb{R}^{r \times n}$
1 **Compute** the SVD of $XX^\top$:
    $[U_s, S, V_s^\top] \leftarrow \text{svd}(XX^\top)$
2 **Compute** $M \leftarrow WU_s S^{1/2}$
3 **Compute** the SVD of $M$:
    $[U, \Sigma, V^\top] \leftarrow \text{svd}(M)$
4 **Let** $U_r, \Sigma_r, V_r = U[:, :r], \Sigma[:r, :r], V_r[:, :r]$
5 **Compute** $S^{-1/2}$
6 **Set** $A \leftarrow U_r$
7 **Set** $B \leftarrow \Sigma_r V_r^\top S^{-1/2} U_s^\top$
8 **return** $A, B$

---

## C   Basics of Low-Rank Approximation

This section presents statements of established results as well as references to their original sources. Although readers may choose to skip this part, it serves to provide greater clarity in the subsequent proofs when referring to these well-known findings.

**Theorem 3.** *(Eckart-Young-Mirsky) Let $A \in \mathbb{R}^{m \times n}$ have the SVD*

$$A = U \Sigma V^\top,$$

*where $\Sigma = \text{diag}(\sigma_1, \sigma_2, \ldots, \sigma_p)$ with $\sigma_1 \geq \sigma_2 \geq \cdots \geq \sigma_p \geq 0$ and $p = \min(m, n)$. For any integer $r$ with $1 \leq r < p$, define the rank-r matrix*

$$A_r = U_r \Sigma_r V_r^\top$$

*by keeping only the top $r$ singular values $\sigma_1, \ldots, \sigma_r$ in $\Sigma$, along with the corresponding columns of $U$ and $V$. Then $A_r$ is a best rank-r approximation to $A$ in the Frobenius norm, i.e.*

$$A_r = \underset{\text{rank}(B) \leq r}{\arg\min} \left\| A - B \right\|_F.$$

*Moreover, if $\sigma_r \neq \sigma_{r+1}$, then this best rank-r approximation $A_r$ is unique.*

*Proof.* See [13]. □

**Corollary 2.** *The solution to the low-rank approximation problem*

$$\min_{\text{rank}(A_k) \leq k} \| A - A_k \|_F,$$

*can be obtained using the formula $A_k = U_k U_k^\top A$ or $A_k = A V_k V_k^\top$, where the SVD of matrix $A$ is given by*

$$A = \begin{bmatrix} U_k & U_k^\perp \end{bmatrix} \begin{bmatrix} \Sigma_k & 0 \\ 0 & \Sigma_k' \end{bmatrix} \begin{bmatrix} V_k & V_k^\perp \end{bmatrix}^\top,$$

**Theorem 4** (Davis-Kahan-Wedin $\sin(\Theta)$ Theorem). *Let $A \in \mathbb{R}^{m \times n}$ be a matrix such that its $r$-th and $(r+1)$-th singular values satisfy $\sigma_r(A) \neq \sigma_{r+1}(A)$. Let $E \in \mathbb{R}^{m \times n}$ be a perturbation matrix, and define $\hat{A} = A + E$. Let $U_r \in \mathbb{R}^{m \times r}$ and $\hat{U}_r \in \mathbb{R}^{m \times r}$ be matrices whose columns consist of the first $r$ left singular vectors of $A$ and $\hat{A}$, respectively. Then,*

$$\left\| U_r U_r^\top - \hat{U}_r \hat{U}_r^\top \right\|_2 \leq \frac{2\|E\|_2}{\sigma_r(A) - \sigma_{r+1}(A)}.$$

*Proof.* This result is proved in "Random perturbation of low rank matrices: Improving classical bounds" [36]. $\qquad\square$

**Lemma 1.** *Let $A \in \mathbb{R}^{d \times r}$ be a rank-$r$ matrix. Then for any $B \in \mathbb{R}^{r \times k}$ it holds that*

$$\sigma_{\min}(A) \, \|B\|_F \;\leq\; \|AB\|_F \;\leq\; \sigma_{\max}(A)\, \|B\|_F = \|A\|_2 \|B\|_F.$$

*Proof.* The proof is classical and can be found, e.g., in [56]. $\qquad\square$

# D  Convergence Proofs for the Full-Rank Regularization Problem

**Theorem 5.** *Let $W \in \mathbb{R}^{m \times n}$ and $X \in \mathbb{R}^{n \times k}$. Suppose that $X$ has full row rank (i.e., $\mathrm{rank}(X) = n$) and that the singular values of $WX$ satisfy $\sigma_r(WX) \neq \sigma_{r+1}(WX)$, where $\sigma_i(\cdot)$ denotes the $i$-th largest singular value. Then, the solution $W_0$ to the problem (1) is unique. Furthermore, if $W_\mu$ denotes the solution to the regularized problem (4), then the following estimate holds:*

$$\|W_0 - W_\mu\|_F \leq \frac{\|W\|_2 \, \|W\|_F}{\sigma_r(WX) - \sigma_{r+1}(WX)} \cdot \frac{\mu}{\sigma_m(X)}$$

*where, $\|\cdot\|_2$ denotes the spectral norm.*

Before we proceed to the proof of Theorem 5, let us establish an auxiliary lemma.

**Lemma 2.** *Let $X \in \mathbb{R}^{m \times n}, m \leq n, \mathrm{rank}(X) = m$. Then*

$$\|(XX^\top)^{1/2} - (XX^\top + \mu I)^{1/2}\|_2 \leq \frac{\mu}{2\sigma_m(X)}.$$

*Proof.* Let $U \Lambda U^\top = XX^\top$ define the eigendecomposition of a symmetric positive definite matrix. Then, $U$ is orthogonal, and the elements of $\Lambda$ are positive.

$$\|(XX^\top)^{1/2} - (XX^\top + \mu I)^{1/2}\|_2 = \|U\Lambda^{1/2}U^\top - (U\Lambda U^\top + \mu UU^\top)^{1/2}\|_2 =$$
$$= \|U(\Lambda^{1/2} - (\Lambda + \mu I)^{1/2})U^\top\|_2 = [\|\cdot\|_2 \text{ is unitarily invariant}] =$$
$$= \|\Lambda^{1/2} - (\Lambda + \mu I)^{1/2}\|_2 = \max_{\lambda \in \sigma(XX^\top)} \left( \sqrt{\lambda + \mu} - \sqrt{\lambda} \right).$$

Note that

$$\sqrt{\lambda + \mu} - \sqrt{\lambda} = \frac{(\sqrt{\lambda + \mu} - \sqrt{\lambda})(\sqrt{\lambda + \mu} + \sqrt{\lambda})}{\sqrt{\lambda + \mu} + \sqrt{\lambda}} = \frac{\mu}{\sqrt{\lambda + \mu} + \sqrt{\lambda}} \leq \frac{\mu}{2\sqrt{\lambda}}.$$

Then

$$\|(XX^\top)^{1/2} - (XX^\top + \mu I)^{1/2}\|_2 = \max_{\lambda \in \sigma(XX^\top)} \left( \sqrt{\lambda + \mu} - \sqrt{\lambda} \right) \leq$$
$$\leq \max_{\lambda \in \sigma(XX^\top)} \frac{\mu}{2\sqrt{\lambda}} = \max_{\sigma \in \sigma(X)} \frac{\mu}{2\sigma} = \frac{\mu}{2\sigma_m(X)}.$$

$\qquad\square$

*Proof of Theorem 5.* The uniqueness of $W_0$ follows from the fact that if $\sigma_r(WX) \neq \sigma_{r+1}(WX)$, then the rank-$r$ low-rank approximation $Y_r$ of the matrix $WX$ is unique (by Eckart-Young-Mirsky Theorem 3). Hence, $W_0$ is a solution if and only if $W_0 X = Y_r$. Moreover, since the kernel of $X$

is empty, if such a matrix $W_0$ exists, it must be unique. However, by Proposition 1 , such a matrix indeed exists.

We now establish the estimate from the theorem's condition. By Proposition 1 , we obtain

$$W_0 = U_0 U_0^\top W,$$

where $U_0$ denotes the first $r$ left singular vectors of $WH_0$, and $H_0 = (XX^\top)^{1/2}$. Analogously, using Proposition 3 , we have

$$W_\mu = U_\mu U_\mu^\top W,$$

where $U_\mu$ denotes the first $r$ left singular vectors of $WH_\mu$, and $H_\mu = (XX^\top + \mu I)^{1/2}$. From Lemma 2 it follows that

$$\|H_0 - H_\mu\|_2 \leq \frac{\mu}{2\,\sigma_m(X)}.$$

Consequently,

$$\|WH_0 - WH_\mu\|_2 \leq \|W\|_2 \|H_0 - H_\mu\|_2 \leq \frac{\|W\|_2}{2\sigma_m(X)}\mu.$$

By applying Davis-Kahan Theorem 4, we obtain

$$\|U_0 U_0^\top - U_\mu U_\mu^\top\|_2 \leq \frac{2\|WH_0 - WH_\mu\|_2}{\sigma_r(WH) - \sigma_{r+1}(WH)} \leq$$

$$\leq \frac{2\|W\|_2}{2\sigma_m(X)\,(\sigma_r(WH) - \sigma_{r+1}(WH))}\mu = \frac{\|W\|_2}{\sigma_m(X)(\sigma_r(WH) - \sigma_{r+1}(WH))}\mu.$$

Thus,

$$\|W_0 - W_\mu\|_F = \|U_0 U_0^\top W - U_\mu U_\mu^\top W\|_F = \|(U_0 U_0^\top - U_\mu U_\mu^\top)W\|_F \leq$$

$$\leq \|U_0 U_0^\top - U_\mu U_\mu^\top\|_2 \|W\|_F \leq \frac{\|W\|_2 \|W\|_F}{\sigma_m(X)(\sigma_r(WH) - \sigma_{r+1}(WH))}\mu.$$

$\square$

# E   Convergence Proofs (Without the Full-Rank Condition)

*Proof of Theorem 1 .*  By Proposition 1 , we obtain

$$W_0 = U_0 U_0^\top W,$$

where $U_0$ denotes the first $r$ left singular vectors of $WX$, and $H_0 = (XX^\top)^{1/2}$. Analogously, using Proposition 3 , we have

$$W_\mu = U_\mu U_\mu^\top W,$$

where $U_\mu$ denotes the first $r$ left singular vectors of $WH_\mu$, and $H_\mu = (XX^\top + \mu I)^{1/2}$. However, we can get the matrices $U_0$ and $U_\mu$ are defined as the matrices of the first $r$ left singular vectors obtained from the singular value decompositions:

$$U_0 \leftarrow \mathrm{SVD}\big(WXX^\top W^\top\big), \qquad U_\mu \leftarrow \mathrm{SVD}\big(W(XX^\top + \mu I)W^\top\big).$$

Here we use the fact that the left singular vectors of a matrix $A$ coincide with the eigenvectors vectors (same that left singular in this case) of $AA^\top$.

Consider the perturbation of the matrix $WXX^\top W^\top$:

$$\|WXX^\top W^\top - W(XX^\top + \mu I)W^\top\|_2 = \mu\|WW^\top\|_2 = \mu\|W\|_2^2.$$

By Applying Davis-Kahan Theorem 4, we obtain

$$\|U_0 U_0^\top - U_\mu U_\mu^\top\|_2 \leq \frac{2\|WXX^\top W^\top - W(XX^\top + \mu I)W^\top\|_2}{\sigma_r(WXX^\top W^\top) - \sigma_{r+1}(WXX^\top W^\top)}.$$

Since $\sigma_k(WXX^\top W^\top) = \sigma_k^2(WX)$, this yields

$$\|U_0 U_0^\top - U_\mu U_\mu^\top\|_2 \le \frac{2\|W\|_2^2}{\sigma_r^2(WX) - \sigma_{r+1}^2(WX)} \mu.$$

Combining this bound with $W_0 = U_0 U_0^\top W$ and $W_\mu = U_\mu U_\mu^\top W$, we arrive at

$$\begin{aligned} \|W_0 - W_\mu\|_F &= \|(U_0 U_0^\top - U_\mu U_\mu^\top)W\|_F \\ &\le \|U_0 U_0^\top - U_\mu U_\mu^\top\|_2 \|W\|_F \\ &\le \frac{2\|W\|_2^2\|W\|_F}{\sigma_r^2(WX) - \sigma_{r+1}^2(WX)} \mu. \end{aligned}$$

$\square$

## F   Implementation Details

Table 5: Choice of hyperparameters for different methods, which were applied to the matrices Q, K, V, O, Up, Down.

| Hyperparameter | LoRA | PiSSA | CorDA | COALA |
|---|---|---|---|---|
| Rank $r$ | 8 | 8 | 8 | 8 |
| $\alpha$ | 12 | 4 | $\frac{1}{2}$ | 8 |
| Dropout | 0.0 | 0.0 | 0.0 | 0.0 |
| Optimizer | AdamW | AdamW | AdamW | AdamW |
| Learning Rate | $1 \times 10^{-4}$ | $1 \times 10^{-4}$ | $1 \times 10^{-4}$ | $1 \times 10^{-4}$ |
| LR Scheduler | Cosine | Cosine | Cosine | Cosine |
| Batch Size | 16 | 16 | 16 | 16 |
| Warmup Steps | 100 | 100 | 100 | 100 |
| Epochs | 1 | 1 | 1 | 1 |

*Fine-tuning:* All training runs were conducted on the same dataset consisting of 40,000 examples, presented in the same order across all experiments. Training a single model required approximately 10 hours, with an additional 2 hours allocated for evaluating the response accuracy on the validation dataset. All experiments were performed on an NVIDIA Tesla T4 GPU with Driver Version 535.183.01 and CUDA Version 12.2. The parameter $\alpha$ was individually selected for each initialization method since the norms resulting from different initialization methods varied, impacting the gradient norms. See Table 5 for the other parameters.

*Compression:* We compressed the Q, K, V, O, Up and Down matrices, approximating each of them with the same rank $r$ to achieve the desired parameter ratio.

## G   Examples

In this section, we present examples supporting various assertions of our work.

**Example G.1** (Loss of Precision When Computing the Gram Matrix [12]). *When "squaring" a matrix and then taking square root, we can lose accuracy in computing its smaller singular values. This phenomenon can be illustrated on the following matrix:*

$$X = \begin{pmatrix} 1 & 1 \\ 0 & \sqrt{\varepsilon} \end{pmatrix},$$

*where $\varepsilon = \varepsilon_m/2$, and $\varepsilon_m$ is a small positive number, representing the machine epsilon (the smallest number such that $1 + \varepsilon_m \ne 1$ in machine arithmetic).*

*First, we compute the singular values of matrix $X$. The singular values are the square roots of the eigenvalues of $X^\top X$:*

$$X^\top X = \begin{pmatrix} 1 & 0 \\ 1 & \sqrt{\varepsilon} \end{pmatrix} \begin{pmatrix} 1 & 1 \\ 0 & \sqrt{\varepsilon} \end{pmatrix} = \begin{pmatrix} 1 & 1 \\ 1 & 1 + \varepsilon \end{pmatrix}.$$

$$\det(X^\top X - \lambda I) = 0.$$

*This leads to:*

$$\lambda^2 - (2 + \varepsilon)\lambda + \varepsilon = 0.$$

$$\lambda = \frac{2 + \varepsilon \pm \sqrt{(2 + \varepsilon)^2 - 4\varepsilon}}{2}.$$

*Thus, the eigenvalues are:*

$$\lambda_1 = \frac{2 + \varepsilon + 2 + \frac{\varepsilon^2}{4}}{2} = 2 + \frac{\varepsilon}{2} + \frac{\varepsilon^2}{8} + \mathcal{O}(\varepsilon^3),$$

$$\lambda_2 = \frac{2 + \varepsilon - \left(2 + \frac{\varepsilon^2}{4}\right)}{2} = \frac{\varepsilon}{2} - \frac{\varepsilon^2}{8} + \mathcal{O}(\varepsilon^3).$$

*The singular values of $X$ are the square roots of the eigenvalues:*

$$\sigma_1 = \sqrt{\lambda_1} = \sqrt{2 + \frac{\varepsilon}{2} + \frac{\varepsilon^2}{8}} = \sqrt{2} \cdot \sqrt{1 + \frac{\varepsilon}{4} + \frac{\varepsilon^2}{16}} = \sqrt{2}\left(1 + \frac{\varepsilon}{8} - \frac{\varepsilon^2}{128}\right) + O(\varepsilon^3).$$

$$\sigma_2 = \sqrt{\lambda_2} = \sqrt{\frac{\varepsilon}{2} - \frac{\varepsilon^2}{8}} = \sqrt{\frac{\varepsilon}{2}} \cdot \sqrt{1 - \frac{\varepsilon}{4}} = \sigma_2 = \frac{\sqrt{\varepsilon}}{\sqrt{2}}\left(1 - \frac{\varepsilon}{8} - \frac{\varepsilon^2}{128}\right) + O(\varepsilon^{3/2}).$$

*Finally,*

$$\sigma_1 = \sqrt{2} + \frac{\sqrt{2}}{8}\varepsilon + O(\varepsilon^2),$$

$$\sigma_2 = \frac{\sqrt{\varepsilon}}{\sqrt{2}} - \frac{\sqrt{\varepsilon}}{8\sqrt{2}}\varepsilon + O(\varepsilon^{3/2}).$$

*However, in machine arithmetic, we will obtain:*

$$XX^\top = \begin{pmatrix} 1 & 1 \\ 1 & 1 \end{pmatrix},$$

*and also*

$$\tilde{\sigma}_1(X) = \sqrt{2}, \quad \tilde{\sigma}_2(X) = 0.$$

*As a result,*

$$|\sigma_2(X) - \tilde{\sigma}_2(X)| = \mathcal{O}(\sqrt{\varepsilon}).$$

*So we lost approximately square root of machine epsilon.*

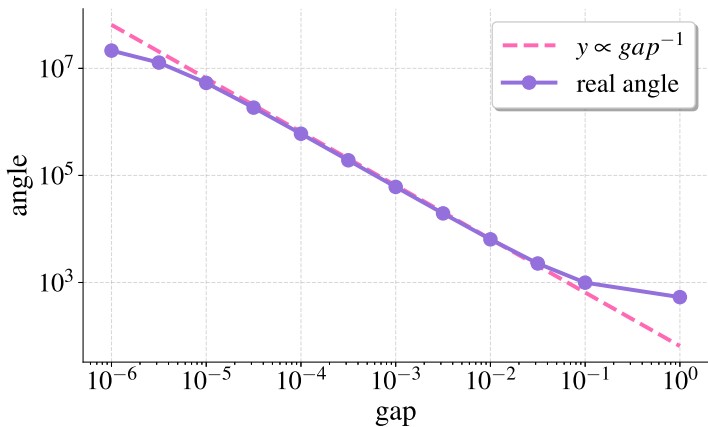

Figure 6: Figure illustrating the dependence of the convergence slope angle of regularized models compared to non-regularized models, with all other factors held constant.

**Example G.2** (Dependence on $gap^{-1}$)**.**

We fixed all dimensional parameters, left and right singular vectors of the matrix $WX$, as well as all singular values except for the $r$-th and $(r+1)$-th ones. Then, we varied the difference between $\sigma_r(WX)$ and $\sigma_{r+1}(WX)$. The convergence rate of the regularized solution to the unregularized solution with respect to this gap is presented Figure 6. We observe that the dependence on the gap is intrinsic to the problem and that we catch the correct asymptotic behaviour in our theoretical bound in the full rank case.

