# OpenReview forum: "COALA: Numerically Stable and Efficient Framework for Context-Aware Low-Rank Approximation"
_NeurIPS.cc/2025/Conference — NeurIPS 2025 poster_

### Official Review · Reviewer_D85U · 2025-06-08

**Clarity:** 3
**Significance:** 3
**Originality:** 4
**Rating:** 5
**Confidence:** 4

**Summary:**

This paper proposes a numerically stable and efficient framework to perform block-wise context-aware low rank approximation on LLMs. Specifically, the paper proposes a QR decomposition-based inverse free optimization algorithm an a regularization to avoid overfitting. Experiments are conducted on multiple LLMs across different tasks.

**Questions:**

Please provide experimental results on the effect of using different $\mu$ in Equation (4)

**Ethical Concerns:**

["NO or VERY MINOR ethics concerns only"]

**Final Justification:**

I'm satisfied with the author's response. After checking all reviews and rebuttals, I'm confident in my judgement that this paper is solid and should be accepted.

**Limitations:**

Yes

**Paper Formatting Concerns:**

No concerns

**Quality:**

3

**Strengths And Weaknesses:**

## Strength
1. The paper provides a novel inversion-free regularized optimization framework to solve the low-rank approximation problem of LLMs
2. The theoretical derivations provided in the paper are clear and solid. The paper has a high writting quality and is easy to follow
3. Adequate experimental results are included to show the improvements compared with baseline methods.

## Weakness
1. Low rank approximation generally leads to a worse performance-efficiency tradeoff compared to pruning and quantization. To this end, though the proposed method improves over existing baseline, it has limited significance on pushing the compression state-of-the-art forward. It is interesting to see if the proposed approximation algorithm or derivations can be used on quantization or pruning as well.
2. The effectiveness of the regularization $\mu$ is worth analyzing. Though the paper provides an theoretical derivation on the impact of $\mu$, an experimental study on different choices of $\mu$ is preferred

---

> ### Author Rebuttal · Authors · 2025-07-31
>
> We sincerely thank you for your time and effort. We appreciate your valuable comments and feedback.
>
> > (1) It is interesting to see if the proposed approximation algorithm or derivations can be used on quantization or pruning as well.
>
> Thank you for this comment. While we acknowledge the important tradeoffs compared to alternative approaches, we believe that our numerically stable framework for weighted low-rank approximation can be useful for developing hybrid compression strategies. For instance, reviewer `FrEc` points out the work [1], which combines quantization and low-rank approximation for compression. This approach also opens up an interesting new research direction for our work.
>
> > (2) Please provide experimental results on the effect of using different $\mu$ in Equation (4).
>
> Thank you, this is indeed an important question to address. The table below presents the results for different compression ratios. Our current findings suggest that different compression ratios within the same model yield comparable in magnitude optimal values of $\mu$, with smaller $\mu$ values tending to correspond to higher compression. As part of our revision, we will expand this analysis with a more comprehensive ablation study on $\mu$ and carefully integrate these insights into the manuscript.
>
> **Table 1.** The dependence of the  _LLaMA-3-8B-Instruct_ model performance on the $\mu$ hyperparameter at different compression ratio values. The average value of the accuracy on the common reasoning dataset is taken as the model's performance.
>
>
> | $\boldsymbol \mu$$\rightarrow $  Ratio $\downarrow $   | **0** | **0.001** | **0.01** | **0.1** | **0.2** | **0.5** | **1** | **2** | **5** | **7** | **10** | **20** | **100** |
> |:------:|:--:|:-----:|:----:|:-:|:--:|:---:|:-:|:---:|:--:|:--:|:--:|:--:|:--:|
> | **90\%** | $59.67_{\color{gray}\pm 0.5}$ | $59.59_{\color{gray}\pm 0.5}$ | $59.94_{\color{gray}\pm 0.5}$ | $60.35_{\color{gray}\pm 0.5}$ | $60.56_{\color{gray}\pm 0.5}$ | $60.76_{\color{gray}\pm 0.5}$| $61.07_{\color{gray}\pm 0.5}$ | $61.47_{\color{gray}\pm 0.5}$ | $61.47_{\color{gray}\pm 0.5}$ | $61.47_{\color{gray}\pm 0.5}$ | $\underline{61.59_{\color{gray}\pm 0.5}}$ | $61.49_{\color{gray}\pm 0.5}$ | $59.80_{\color{gray}\pm 0.4}$
> | **70\%**   | $45.39_{\color{gray}\pm0.4}$ | $45.44_{\color{gray}\pm0.4}$ | $45.75_{\color{gray}\pm0.4}$ | $46.27_{\color{gray}\pm0.4}$ | $46.18_{\color{gray}\pm0.4}$ | $46.67_{\color{gray}\pm0.4}$ | $47.13_{\color{gray}\pm0.5}$ | $47.25_{\color{gray}\pm0.5}$ | $\underline{47.54_{\color{gray}\pm0.5}}$ | $47.44_{\color{gray}\pm0.5}$ | $47.16_{\color{gray}\pm0.5}$ | $46.68_{\color{gray}\pm0.4}$ | $41.00_{\color{gray}\pm0.4}$
> | **60\%** | $41.09_{\color{gray}\pm 0.4}$ | $41.04_{\color{gray}\pm 0.4}$ | $41.23_{\color{gray}\pm 0.4}$  | $41.48_{\color{gray}\pm 0.4}$ | $41.62_{\color{gray}\pm 0.4}$ | $41.76_{\color{gray}\pm 0.4}$ | $\underline{41.88_{\color{gray}\pm 0.5}}$ | $41.85_{\color{gray}\pm 0.5}$ | $41.39_{\color{gray}\pm 0.5}$ | $41.18_{\color{gray}\pm 0.5}$ | $40.60_{\color{gray}\pm 0.5}$ | $39.50_{\color{gray}\pm 0.4}$ | $36.25_{\color{gray}\pm0.4}$
> | **50\%** | $37.12_{\color{gray}\pm 0.4}$ | $37.2_{\color{gray}\pm 0.4}$ | $37.28_{\color{gray}\pm 0.4}$ | $37.77_{\color{gray}\pm 0.4}$ | $\underline{37.90_{\color{gray}\pm 0.4}}$ | $37.51_{\color{gray}\pm 0.4}$ | $37.26_{\color{gray}\pm 0.4}$ | $37.15_{\color{gray}\pm 0.4}$ | $37.15_{\color{gray}\pm 0.4}$ | $37.22_{\color{gray}\pm 0.4}$ | $36.64_{\color{gray}\pm 0.4}$ | $36.18_{\color{gray}\pm 0.4}$  | $35.29_{\color{gray}\pm 0.4}$
>
> ___
>
> [1] Saha, Rajarshi, Naomi Sagan, Varun Srivastava, Andrea Goldsmith, and Mert Pilanci. "Compressing large language models using low rank and low precision decomposition." Advances in Neural Information Processing Systems 37 (2024): 88981-89018.

---

> > ### Comment · Reviewer_D85U · 2025-08-03
> >
> > I would like to thank the author for the additional information. I'm satisfied with the response.

---

### Official Review · Reviewer_FrEc · 2025-06-25

**Clarity:** 2
**Significance:** 3
**Originality:** 2
**Rating:** 5
**Confidence:** 4

**Summary:**

This paper introduces CoTAn -- a numerically stable and inversion free framework to minimize the optimization problem $$\text{minimize}_{W' : \text{rank}(W') \leq r} ||WX - W'X||_F,$$ where $X$ is a calibration data matrix. This problem is referred to as the *context-aware low-rank approximation* problem in the paper. Unlike prior approaches mentioned (e.g., SVD-LLM, ASVD), CoTAn avoids computing and inverting the Gram matrix of the calibration data, i.e., $X^\top X$, which is often ill-conditioned or singular in practice. Hence, the name, "numerically stable" in the title of the paper.

This is done using stable matrix factorizations such as QR and SVD decompositions directly on the transformed input data (rather than the Gram matrix) -- optionally combined with regularization to handle ill-posed or underdetermined settings. The paper also discusses an efficient Tall-Skinny QR decomposition (TSQR) for scalability to large matrices, and proves convergence and error bounds for the regularized variant. Experiments on Llama-3 family of models and Mistral, and 8 downstream evaluation tasks (BoolQ, PIQA, SIQA, HellaSwag, Winogrande, ARC-e, ARC-c, OBQA) show that CoTAn achieves comparable or better performance with greater stability and faster runtime, when compared to ASVD and SVD-LLM.

**Questions:**

I have some questions and would appreciate it if they are addressed:

1. Can the authors comment on how sensitive CoTAN is to the choice of the regularization parameter $\mu$? Is there a principled way to choose it (for example, via calibration)? Such ablation studies would be helpful in understanding the role of a proximal regularizer that penalizes deviation, such as in (4).

2. Any ablation study with the choice of rank?

Please also refer to other important concerns in the previous strengths and weaknesses section.

**Ethical Concerns:**

["NO or VERY MINOR ethics concerns only"]

**Final Justification:**

I have raised my score in light of the author's rebuttal, which is satisfactory.

**Limitations:**

Yes. Would appreciate if authors would comment of the limitation that low-rank approximation by itself leads to significant accuracy drop.

**Paper Formatting Concerns:**

None.

**Quality:**

2

**Strengths And Weaknesses:**

**Strengths**:
The paper address a relevant problem in the field of LLM compression -- for compressing weight matrices in an LLM, it is not sufficient to preserve just the Frobenius norm error of the weight matrix, but rather the output activations of each layer. Consequently, several LLM compression methods have resorted to using a small calibration dataset for this purpose, and this paper studies low-rank approximation with a calibration dataset. The paper addresses a practical issue regarding matrix inversion errors while inverting (near) singular Gram matrices, while providing clean mathematical formulations, correctness proofs, and convergence guarantees for the same. Numerical experiments also validate the theoretical claims.

In terms of clarity, the paper, in general, seems well-organized and technically sound with a clear motivation for each design choice. The pseudocode and figures also help in understanding and are highly appreciated. Some important derivations are delegated to the Appendix, but could benefit from an intuitive explanation in the main text.

**Weakness**:
I believe the paper can do a better job placing itself in the context of some relevant works. For instance, [this paper](https://proceedings.neurips.cc/paper_files/paper/2024/hash/a20e8451ffb07ad25282c21945ad4f19-Abstract-Conference.html) is quite relevant and adopts a similar approach to solve the context-aware low-rank approximation problem. How does CoTAn compare with the approach presented in this paper? For instance, Lemma 4.2 (and its complete derivation in Appendix B) of this paper, also solves the same problem, without inverting any singular matrix. The rank-constrained regression problem (eq. 5 in this paper ) seems to be the same problem as CoTAn solves (up to some change in variables). The authors seem to have missed discussing this in related works.

Moreover, low-rank decomposition of the weight matrix, by itself, introduces a very large gap in downstream accuracies with respect to the uncompressed baselines. This is true for all methods like ASVD, SVD-LLM and including CoTAn. This often renders low-rank factorizations not-so-useful in practical application where accuracy requirements are paramount. Some discussion needs to be added about how CoTAn is compatible with other methods that can help reduce the accuracy gap between low-rank factorized and uncompressed models.

I would be happy to revise my score if my concerns are addressed.

---

> ### Author Rebuttal · Authors · 2025-07-31
>
> We sincerely thank you for your time and effort. We appreciate your valuable comments and feedback.
>
>
> > (1) How does CoTAn compare with the approach presented in this paper? For instance, Lemma 4.2 (and its complete derivation in Appendix B) of this paper, also solves the same problem, without inverting any singular matrix.
>
> Thank for poitining out this interesting work. We will definitely add this study to our literature review. We note that in Lemma 4.2 the authors still invert singular values of the matrix $X$, which can numerically be very close to zeroes (see, for example, Figure 2 in our work):
> $$ Z_\star \triangleq \mathop{\operatorname*{argmin}}_{\operatorname{rank}( Z)\le k} \|\|  X Z- Y \|\|_F^{2} = ( V  I_m \boldsymbol \Sigma^{-1}  {\acute{U}}  I_k) ( I_k^{\top} {\acute{\Sigma}} {\acute{V}}^{\top}),$$
> where $\boldsymbol \Sigma := {\tilde \Sigma } I_m \in \mathbb{R}^{m \times m}$
> is a diagonal matrix consisting of the non-zero singular values of $X$. At the same time our approach is fully free from any matrix inverse.
>
> > (2) Can the authors comment on how sensitive CoTAN is to the choice of the regularization parameter $\mu$? Is there a principled way to choose it (for example, via calibration)? Such ablation studies would be helpful in understanding the role of a proximal regularizer that penalizes deviation, such as in (4).
>
>
> We thank the reviewer for this valuable suggestion.
> The table below presents the results for different compression ratios. Our current findings suggest that different compression ratios within the same model yield comparable in magnitude optimal values of $\mu$, with smaller $\mu$ values tending to correspond to higher compression. As part of our revision, we will expand this analysis with a more comprehensive ablation study on $\mu$ and carefully integrate these insights into the manuscript.
>
>
> **Table 1.** The dependence of the  _LLaMA-3-8B-Instruct_ model performance on the $\mu$ hyperparameter at different compression ratio values. The average value of the accuracy on the common reasoning dataset is taken as the model's performance.
>
>
> | $\boldsymbol \mu$$\rightarrow $  Ratio $\downarrow $   | **0** | **0.001** | **0.01** | **0.1** | **0.2** | **0.5** | **1** | **2** | **5** | **7** | **10** | **20** | **100** |
> |:------:|:--:|:-----:|:----:|:-:|:--:|:---:|:-:|:---:|:--:|:--:|:--:|:--:|:--:|
> | **90\%** | $59.67_{\color{gray}\pm 0.5}$ | $59.59_{\color{gray}\pm 0.5}$ | $59.94_{\color{gray}\pm 0.5}$ | $60.35_{\color{gray}\pm 0.5}$ | $60.56_{\color{gray}\pm 0.5}$ | $60.76_{\color{gray}\pm 0.5}$| $61.07_{\color{gray}\pm 0.5}$ | $61.47_{\color{gray}\pm 0.5}$ | $61.47_{\color{gray}\pm 0.5}$ | $61.47_{\color{gray}\pm 0.5}$ | $\underline{61.59_{\color{gray}\pm 0.5}}$ | $61.49_{\color{gray}\pm 0.5}$ | $59.80_{\color{gray}\pm 0.4}$
> | **70\%**   | $45.39_{\color{gray}\pm0.4}$ | $45.44_{\color{gray}\pm0.4}$ | $45.75_{\color{gray}\pm0.4}$ | $46.27_{\color{gray}\pm0.4}$ | $46.18_{\color{gray}\pm0.4}$ | $46.67_{\color{gray}\pm0.4}$ | $47.13_{\color{gray}\pm0.5}$ | $47.25_{\color{gray}\pm0.5}$ | $\underline{47.54_{\color{gray}\pm0.5}}$ | $47.44_{\color{gray}\pm0.5}$ | $47.16_{\color{gray}\pm0.5}$ | $46.68_{\color{gray}\pm0.4}$ | $41.00_{\color{gray}\pm0.4}$
> | **60\%** | $41.09_{\color{gray}\pm 0.4}$ | $41.04_{\color{gray}\pm 0.4}$ | $41.23_{\color{gray}\pm 0.4}$  | $41.48_{\color{gray}\pm 0.4}$ | $41.62_{\color{gray}\pm 0.4}$ | $41.76_{\color{gray}\pm 0.4}$ | $\underline{41.88_{\color{gray}\pm 0.5}}$ | $41.85_{\color{gray}\pm 0.5}$ | $41.39_{\color{gray}\pm 0.5}$ | $41.18_{\color{gray}\pm 0.5}$ | $40.60_{\color{gray}\pm 0.5}$ | $39.50_{\color{gray}\pm 0.4}$ | $36.25_{\color{gray}\pm0.4}$
> | **50\%** | $37.12_{\color{gray}\pm 0.4}$ | $37.2_{\color{gray}\pm 0.4}$ | $37.28_{\color{gray}\pm 0.4}$ | $37.77_{\color{gray}\pm 0.4}$ | $\underline{37.90_{\color{gray}\pm 0.4}}$ | $37.51_{\color{gray}\pm 0.4}$ | $37.26_{\color{gray}\pm 0.4}$ | $37.15_{\color{gray}\pm 0.4}$ | $37.15_{\color{gray}\pm 0.4}$ | $37.22_{\color{gray}\pm 0.4}$ | $36.64_{\color{gray}\pm 0.4}$ | $36.18_{\color{gray}\pm 0.4}$  | $35.29_{\color{gray}\pm 0.4}$
>
>
> As for the principled way to choose it, we naturally observe that for different models the parameter $\mu$ shows dependence on the scales of $W$ and $X$ (as is also supported by our theoretical estimates). The matrix norms can be computed prior to the compression step and as part of our revision, we will expand this analysis with more comprehensive ablation studies on $\mu$ and carefully incorporate these findings in the manuscript.
>
> > (3) Any ablation study with the choice of rank?
>
> We thank the reviewer for this question. Our experiments examining the relationship between compression (for each layer rank is determined by the compression ratio) and optimal $\mu$ values revealed that lower compression ratios corresponded to smaller optimal $\mu$ values. This observation aligns with the intuition that reduced parameter counts (from lower ranks) may decrease susceptibility to overfitting on calibration data. Hence, regularization is less necessary in these cases. Please see the table with optimal $\mu$ below:
>
> **Table 2.** The dependence of the optimal $\mu$ on compressing ratio.
>
> | **Ratio, \%** | 90 | 70 | 60 | 50 |
> |:----------------:|:---------:|:--:|:--:|:--:|
> |**Optimal $\boldsymbol \mu \boldsymbol \approx$**| 10 | 5 | 1 | 0.2 |

---

### Official Review · Reviewer_kPsW · 2025-07-02

**Clarity:** 4
**Significance:** 4
**Originality:** 4
**Rating:** 6
**Confidence:** 4

**Summary:**

The authors proposes an approach based on decompositions that is numerically stable compared to methods that relies on Gram matrix and their subsequent inversion. CoTAn introduces an inversion-free solution based on stable matrix decompositions like QR and SVD. The framework is comprehensive, proposing the TSQR algorithm for memory-constrained scenarios and a novel regularization technique for data-scarce settings. The author shows that the method is appliable on large language models namely LLama 3.1 and Mistral 7B and is compared on wide datasets, demonstrating superior performance and stability in compression compared to existing methods.

**Questions:**

*  To improve the practical utility of the method, could you provide a sensitivity analysis or a plot showing the effect of different µ values on model performance? This would offer valuable guidance for selecting this hyperparameter.
*  Adaptive rank to compare the methods fairly though it is an important aspect for compression ( L225 - 226) could you please either showcase the overhead those methods adds. Models are compressed once and then  deployed, if the margin in performance is large sometimes it is worth the effort to tune

**Ethical Concerns:**

["NO or VERY MINOR ethics concerns only"]

**Final Justification:**

I stay with my recommendation of accepting the paper. The authors answered my concerns

**Limitations:**

Yes

**Quality:**

4

**Strengths And Weaknesses:**

### Strengths:

* Tackles a Critical Problem: The paper addresses the crucial and practical issue of numerical stability during the compression and fine-tuning of large language models, a significant bottleneck for current methods.

* Comprehensive and Versatile Framework: The proposed solution is not a single trick but a complete framework. The extension to handle memory-constrained scenarios via TSQR and data-scarce situations via regularization makes CoTAn robust and widely applicable.

* Diversified and Strong Evaluation: The method is thoroughly evaluated on multiple modern architectures (LLaMA3-8B, LLaMA3-1B, Mistral-7B) across a wide range of  datasets, providing convincing evidence of its effectiveness. The work also clearly shows how to choose between SVD and QR decomposition for efficiency depending on the hardware and matrix dimensions


* Solid Theoretical and Practical Support: The claims are well-supported by clear mathematical propositions and proofs. Furthermore, the inclusion of pseudocode and detailed hyperparameters in the appendix significantly aids reproducibility.


### Weakness :
The following weaknesses are minor and more for readability
* Guidance on Regularization Parameter: The method's effectiveness is clearly shown, yet the sensitivity of the performance to the regularization parameter µ is not fully explored. For instance, the paper states with each results the value of μ ,
 > CoTAnμ​ is regularized approach, we get µ = 5 · 10−4", but the link between this choice and the final accuracy is not established."

---

> ### Author Rebuttal · Authors · 2025-07-31
>
> We sincerely thank you for your time and effort. We appreciate your valuable comments and feedback.
>
> > (1) To improve the practical utility of the method, could you provide a sensitivity analysis or a plot showing the effect of different $\mu$ values on model performance? This would offer valuable guidance for selecting this hyperparameter.
>
> We appreciate this question. The table below presents the results for different compression ratios. Our current findings suggest that different compression ratios within the same model yield comparable in magnitude optimal values of $\mu$, with smaller $\mu$ values tending to correspond to higher compression.  As part of our revision, we will expand this analysis with a more comprehensive ablation study on $\mu$ and carefully integrate these insights into the manuscript.
>
> **Table 1.** The dependence of the  _LLaMA-3-8B-Instruct_ model performance on the $\mu$ hyperparameter at different compression ratio values. The average value of the accuracy on the common reasoning dataset is taken as the model's performance.
>
>
> | $\boldsymbol \mu$$\rightarrow $  Ratio $\downarrow $   | **0** | **0.001** | **0.01** | **0.1** | **0.2** | **0.5** | **1** | **2** | **5** | **7** | **10** | **20** | **100** |
> |:------:|:--:|:-----:|:----:|:-:|:--:|:---:|:-:|:---:|:--:|:--:|:--:|:--:|:--:|
> | **90\%** | $59.67_{\color{gray}\pm 0.5}$ | $59.59_{\color{gray}\pm 0.5}$ | $59.94_{\color{gray}\pm 0.5}$ | $60.35_{\color{gray}\pm 0.5}$ | $60.56_{\color{gray}\pm 0.5}$ | $60.76_{\color{gray}\pm 0.5}$| $61.07_{\color{gray}\pm 0.5}$ | $61.47_{\color{gray}\pm 0.5}$ | $61.47_{\color{gray}\pm 0.5}$ | $61.47_{\color{gray}\pm 0.5}$ | $\underline{61.59_{\color{gray}\pm 0.5}}$ | $61.49_{\color{gray}\pm 0.5}$ | $59.80_{\color{gray}\pm 0.4}$
> | **70\%**   | $45.39_{\color{gray}\pm0.4}$ | $45.44_{\color{gray}\pm0.4}$ | $45.75_{\color{gray}\pm0.4}$ | $46.27_{\color{gray}\pm0.4}$ | $46.18_{\color{gray}\pm0.4}$ | $46.67_{\color{gray}\pm0.4}$ | $47.13_{\color{gray}\pm0.5}$ | $47.25_{\color{gray}\pm0.5}$ | $\underline{47.54_{\color{gray}\pm0.5}}$ | $47.44_{\color{gray}\pm0.5}$ | $47.16_{\color{gray}\pm0.5}$ | $46.68_{\color{gray}\pm0.4}$ | $41.00_{\color{gray}\pm0.4}$
> | **60\%** | $41.09_{\color{gray}\pm 0.4}$ | $41.04_{\color{gray}\pm 0.4}$ | $41.23_{\color{gray}\pm 0.4}$  | $41.48_{\color{gray}\pm 0.4}$ | $41.62_{\color{gray}\pm 0.4}$ | $41.76_{\color{gray}\pm 0.4}$ | $\underline{41.88_{\color{gray}\pm 0.5}}$ | $41.85_{\color{gray}\pm 0.5}$ | $41.39_{\color{gray}\pm 0.5}$ | $41.18_{\color{gray}\pm 0.5}$ | $40.60_{\color{gray}\pm 0.5}$ | $39.50_{\color{gray}\pm 0.4}$ | $36.25_{\color{gray}\pm0.4}$
> | **50\%** | $37.12_{\color{gray}\pm 0.4}$ | $37.2_{\color{gray}\pm 0.4}$ | $37.28_{\color{gray}\pm 0.4}$ | $37.77_{\color{gray}\pm 0.4}$ | $\underline{37.90_{\color{gray}\pm 0.4}}$ | $37.51_{\color{gray}\pm 0.4}$ | $37.26_{\color{gray}\pm 0.4}$ | $37.15_{\color{gray}\pm 0.4}$ | $37.15_{\color{gray}\pm 0.4}$ | $37.22_{\color{gray}\pm 0.4}$ | $36.64_{\color{gray}\pm 0.4}$ | $36.18_{\color{gray}\pm 0.4}$  | $35.29_{\color{gray}\pm 0.4}$
>
> As for the different models, we observe that they exhibit distinct scaling properties in the matrices $W$ and $X$, which directly influences the optimal choice of $\mu$. While our scale-dependent heuristics produce $\mu$ values that improve model performance, we find that further tuning yields additional gains. We will also be sure to study this aspect in more detail and conduct more thorough ablation studies regarding $\mu$ in our revised manuscript.
>
>
> > (2) Adaptive rank to compare the methods fairly though it is an important aspect for compression (L225 - 226) could you please either showcase the overhead those methods adds. Models are compressed once and then deployed, if the margin in performance is large sometimes it is worth the effort to tune.
>
> We agree that adaptive ranking is an important aspect of compression. Unfortunately, there is no definitive answer regarding the overhead that these methods introduce, as they vary significantly and may require different computational times. For example, the method from work [1] emperically chooses rank based on error approximation in the Frobenius norm, so the choice of rank is not a computationally demandning step. On the other hand, work [2] presents a differentiable method to solve for optimal truncation positions, which requires 8 GPU hours for the LLama-7B model. In our work, we wanted to study how different algorithms for compressing matrices affect quality, so we eliminated the interfering factor using a single shared strategy. Nevertheless, as you noted, while the extended initialization time due to rank adaptation may be justified by the model's improved deployment performance, combining our method with these techniques can  certainly be beneficial. We plan to explore optimal strategies in future work.
> ___
> [1] Xin Wang, Yu Zheng, Zhongwei Wan, and Mi Zhang. "SVD-LLM: Truncation-aware Singular Value Decomposition for Large Language Model Compression." . In The Thirteenth International Conference on Learning Representations (2025).
>
> [2] Wang Qinsi, Jinghan Ke, Masayoshi Tomizuka, Kurt Keutzer, and Chenfeng Xu. "Dobi-SVD: Differentiable SVD for LLM Compression and Some New Perspectives." . In The Thirteenth International Conference on Learning Representations (2025).

---

> > ### Comment · Reviewer_kPsW · 2025-08-04
> >
> > Thank you for the additional experiment. Overall a very solid paper/ well written with practical contribution and easy integration. I stay with my rating.

---

### Official Review · Reviewer_GaTG · 2025-07-03

**Clarity:** 3
**Significance:** 2
**Originality:** 3
**Rating:** 4
**Confidence:** 4

**Summary:**

CoTAn aims to preserve the most important directions of the activations. The method computes a low-rank decomposition that approximately minimizes error on the layer’s output activations, rather than weight reconstruction error itself. To achieve this efficiently, CoTAn uses an inversion-free algorithm: instead of explicitly inverting large covariance matrices or solving normal equations, it leverages orthogonal projections and QR factorizations to find the low-dimensional subspace spanned by the data, and then performs a truncated SVD or low-rank factorization in that subspace. This approach avoids  matrix inverses and can be implemented in a distributed manner.

**Questions:**

- What happens when the context data used for compression is misaligned with test-time data—how robust is the method?
- Can the authors clarify how CoTAn would be integrated into a full training or fine-tuning pipeline, or is it solely for post-training compression

**Ethical Concerns:**

["NO or VERY MINOR ethics concerns only"]

**Final Justification:**

During the rebuttal the authors  addressed my questions  on  misaligned data, as well as method applicability for both post-training and fine-tuning scenarios, thus I intend to raise my score.

**Limitations:**

Yes

**Paper Formatting Concerns:**

No major issues

**Quality:**

3

**Strengths And Weaknesses:**

*Strengths*
- The paper addresses a meaningful problem in model compression: numerical instability in context-aware low-rank approximations. The authors propose an inversion-free formulation that avoids inverting Gram matrices, the use of QR and TSQR decompositions is well-motivated from numerical linear algebra.

*Weaknesses*

- Related literature is not well addressed. Statement "Structured pruning removes unnecessary parameters and simplifies the
model architecture without significant loss of accuracy In turn, the low rank decomposition approach is often memory-efficient and can accelerate model inference" is not correct in general,  compression rate/accuracy drop depends significantly on the data, neural network architecture, way of training, tensor decompositions sometimes can lead to slow down of inference if rank is not small enough or due do increased number of reading-writing accesses to the memory.
- The paper's  claims like stability and  generalization are not well demonstrated by empirical results

---

> ### Author Rebuttal · Authors · 2025-07-31
>
> We sincerely thank you for your time and effort. We appreciate your valuable comments and feedback.
>
>
>
> > (1) Related literature is not well addressed. Statement "...In turn, the low rank decomposition approach is often memory-efficient and can accelerate model inference ..." is not correct in general, compression rate/accuracy drop depends significantly on the data, neural network architecture, way of training, tensor decompositions sometimes can lead to slow down of inference if rank is not small enough or due do increased number of reading-writing accesses to the memory.
>
> We do agree that the efficiency/accuracy trade-off of any compression technique is highly problem-dependent, and also that more more complex structures such as tensor decompositions can additionally slow down computations. We will be sure to highlight these points and also incorporate suggestions from the other reviewers in the related work section to enhance its correctness and completeness.
>
> > (2) What happens when the context data used for compression is misaligned with testtime data -- how robust is the method?
>
> Thank you for this question, it is indeed an important aspect to consider. On the one hand, we are in the setting of weighted matrix approximation for compression and our method inherits its properties. On the other hand, we have an additional regularization term. This term serves to penalize deviations from the original (unweighted) matrix, mitigating this problem and, hence, should positvely affect generalization. In the table below we present results with misaligned testtime data. We observe that the regularization parameter does increase the metrics.
>
>
> **Table. 1**  The dependence of the  _LLaMA-3-1B_ model performance on the $\mu$ hyperparameter at a compression ratio 90\%, using Wiki2 as compression dataset. The average value of the accuracy on the common reasoning dataset is taken as the model's performance.
>
> | **$\boldsymbol \mu$** | 0 | 0.01 | 0.1 | 0.5 | 2 | 5 | 10 | 20 | 50 | 100 |
> |:---------:|:---:|:-----:|:----:|:----:|:----:|:--:|:--:|:---:|:--:|:--:|
> | **Avg.**  | $41.08{\color{gray}\small\pm0.4}$ | $41.15{\color{gray}\small\pm0.4}$ | $41.37{\color{gray}\small\pm0.4}$ | $41.41{\color{gray}\small\pm0.4}$ | $41.61{\color{gray}\small\pm0.4}$ | $42.01{\color{gray}\small\pm0.4}$ | $\underline{42.04}{\color{gray}\small\pm0.4}$ | $42.03{\color{gray}\small\pm0.4}$ | $40.91{\color{gray}\small\pm0.4}$ | $39.59{\color{gray}\small\pm0.4}$ |
>
>
> > (3) Can the authors clarify how CoTAn would be integrated into a full training or finetuning pipeline, or is it solely for post-training compression?
>
> Thank you for your question. We focus on two settings involving weighted low-rank approximation. Beyond post-training compression (Section 6.1), we also demonstrate its use in constructing initializations for fine-tuning (Section 6.2). Due to the generality of our framework, we presume that it can be also useful in other scenarios where weighted low-rank approximation is of interest. Furthermore, it could be combined with other techniques — for example, by integrating different training methods after post-training compression. We consider this an interesting research direction and plan to investigate it further in future work.
>
> As for the full training procedures, the potential usefulness of weighted low-rank approximation remains unclear for us at this stage. While it might be relevant for low-rank gradient approximations (e.g., as in GaLore), this is speculative rather than empirically validated.
>
> ___
> ### Additional experiments
>
> We would also like to highlight that in addition to addressing your specific questions, we have included the results of a new experiment responding to a common concern raised by several reviewers. Specifically, multiple reviewers requested an analysis of the parameter $\mu$ dependence, which we present in the table below.
>
> **Table 1.** The dependence of the  _LLaMA-3-8B-Instruct_ model performance on the $\mu$ hyperparameter at different compression ratio values. The average value of the accuracy on the common reasoning dataset is taken as the model's performance.
>
>
> | $\boldsymbol \mu$$\rightarrow $  Ratio $\downarrow $   | **0** | **0.001** | **0.01** | **0.1** | **0.2** | **0.5** | **1** | **2** | **5** | **7** | **10** | **20** | **100** |
> |:------:|:--:|:-----:|:----:|:-:|:--:|:---:|:-:|:---:|:--:|:--:|:--:|:--:|:--:|
> | **90\%** | $59.67_{\color{gray}\pm 0.5}$ | $59.59_{\color{gray}\pm 0.5}$ | $59.94_{\color{gray}\pm 0.5}$ | $60.35_{\color{gray}\pm 0.5}$ | $60.56_{\color{gray}\pm 0.5}$ | $60.76_{\color{gray}\pm 0.5}$| $61.07_{\color{gray}\pm 0.5}$ | $61.47_{\color{gray}\pm 0.5}$ | $61.47_{\color{gray}\pm 0.5}$ | $61.47_{\color{gray}\pm 0.5}$ | $\underline{61.59_{\color{gray}\pm 0.5}}$ | $61.49_{\color{gray}\pm 0.5}$ | $59.80_{\color{gray}\pm 0.4}$
> | **70\%**   | $45.39_{\color{gray}\pm0.4}$ | $45.44_{\color{gray}\pm0.4}$ | $45.75_{\color{gray}\pm0.4}$ | $46.27_{\color{gray}\pm0.4}$ | $46.18_{\color{gray}\pm0.4}$ | $46.67_{\color{gray}\pm0.4}$ | $47.13_{\color{gray}\pm0.5}$ | $47.25_{\color{gray}\pm0.5}$ | $\underline{47.54_{\color{gray}\pm0.5}}$ | $47.44_{\color{gray}\pm0.5}$ | $47.16_{\color{gray}\pm0.5}$ | $46.68_{\color{gray}\pm0.4}$ | $41.00_{\color{gray}\pm0.4}$
> | **60\%** | $41.09_{\color{gray}\pm 0.4}$ | $41.04_{\color{gray}\pm 0.4}$ | $41.23_{\color{gray}\pm 0.4}$  | $41.48_{\color{gray}\pm 0.4}$ | $41.62_{\color{gray}\pm 0.4}$ | $41.76_{\color{gray}\pm 0.4}$ | $\underline{41.88_{\color{gray}\pm 0.5}}$ | $41.85_{\color{gray}\pm 0.5}$ | $41.39_{\color{gray}\pm 0.5}$ | $41.18_{\color{gray}\pm 0.5}$ | $40.60_{\color{gray}\pm 0.5}$ | $39.50_{\color{gray}\pm 0.4}$ | $36.25_{\color{gray}\pm0.4}$
> | **50\%** | $37.12_{\color{gray}\pm 0.4}$ | $37.2_{\color{gray}\pm 0.4}$ | $37.28_{\color{gray}\pm 0.4}$ | $37.77_{\color{gray}\pm 0.4}$ | $\underline{37.90_{\color{gray}\pm 0.4}}$ | $37.51_{\color{gray}\pm 0.4}$ | $37.26_{\color{gray}\pm 0.4}$ | $37.15_{\color{gray}\pm 0.4}$ | $37.15_{\color{gray}\pm 0.4}$ | $37.22_{\color{gray}\pm 0.4}$ | $36.64_{\color{gray}\pm 0.4}$ | $36.18_{\color{gray}\pm 0.4}$  | $35.29_{\color{gray}\pm 0.4}$

---

> > ### Comment · Reviewer_GaTG · 2025-08-07
> >
> > Dear Authors,
> >
> > Thank you for the response. It’s good that you plan to include discussions on efficiency/accuracy tradeoffs to improve the related work section. The explanation and results about misaligned data make the method more convincing. It’s also helpful to know CoTAn can be used for both compression and constructing initializations for fine-tuning . Also thank you for sharing  more experiments on $\mu$  hyperparameter, they are helpful.  Because of these improvements, I intend to  raise my score.
> >
> > Best,
> > Reviewer

---

### Decision · Program_Chairs · 2025-09-17

**Decision:**

Accept (poster)

**Comment:**

This work addresses the inversion of gram matrices in context aware low rank approximation. It is based on the observation that these matrices are highly ill-conditioned and thus, following conventional wisdom in numerical linear algebra, the formation of these matrices should be avoided to preserve numerical accuracy. The authors propose using classical techniques from numerical linear algebra, such as QR factorization and the TSQR algorithm that instead operate on the square root of the gram matrix. The authors provide extensive numerical evidence that this modification yields moderate but significant improvements in practice. The reviewers commend the clarity of the proposed insight and the thoroughness of the evaluation. While it was pointed out that the improvements could be more striking, the paper seems to bring a valuable novel perspective to the compression and fine-tuning of large language models. Thus, I recommend acceptance